# Bacteriophage Cocktails in the Post-COVID Rehabilitation

**DOI:** 10.3390/v14122614

**Published:** 2022-11-23

**Authors:** Fedor M. Zurabov, Ekaterina A. Chernevskaya, Natalia V. Beloborodova, Alexander Yu. Zurabov, Marina V. Petrova, Mikhail Ya. Yadgarov, Valentina M. Popova, Oleg E. Fatuev, Vladislav E. Zakharchenko, Marina M. Gurkova, Ekaterina A. Sorokina, Egor A. Glazunov, Tatiana A. Kochetova, Victoria V. Uskevich, Artem N. Kuzovlev, Andrey V. Grechko

**Affiliations:** 1Research and Production Center “MicroMir”, 5/23 Nizhny Kiselny Lane, bldg 1, 107031 Moscow, Russia; 2Federal Research and Clinical Center of Intensive Care Medicine and Rehabilitology, 25 Petrovka Str., 2 bldg, 10703 Moscow, Russia

**Keywords:** rehabilitation, COVID-19, bacteriophages, phage cocktail, phage therapy, gut dysbiosis, microbiome, microbiota, *Klebsiella pneumoniae*, 3D Cell Explorer

## Abstract

Increasing evidence suggests that gut dysbiosis is associated with coronavirus disease 2019 (COVID-19) infection and may persist long after disease resolution. The excessive use of antimicrobials in patients with COVID-19 can lead to additional destruction of the microbiota, as well as to the growth and spread of antimicrobial resistance. The problem of bacterial resistance to antibiotics encourages the search for alternative methods of limiting bacterial growth and restoring the normal balance of the microbiota in the human body. Bacteriophages are promising candidates as potential regulators of the microbiota. In the present study, two complex phage cocktails targeting multiple bacterial species were used in the rehabilitation of thirty patients after COVID-19, and the effectiveness of the bacteriophages against the clinical strain of *Klebsiella pneumoniae* was evaluated for the first time using real-time visualization on a 3D Cell Explorer microscope. Application of phage cocktails for two weeks showed safety and the absence of adverse effects. An almost threefold statistically significant decrease in the anaerobic imbalance ratio, together with an erythrocyte sedimentation rate (ESR), was detected. This work will serve as a starting point for a broader and more detailed study of the use of phages and their effects on the microbiome.

## 1. Introduction

Gut health and gut microbiota are increasingly associated with a variety of chronic diseases. Imbalances in the gut microbiota caused by poor diet, stress, antibiotic use, and other lifestyle and environmental factors are associated with the development of intestinal inflammation and gut disorders. Scientists are finding more and more links between gut microbiota and human health, from associations with autoimmune diseases and metabolic conditions to mental health and effects on brain function [1,2].

Increasing evidence suggests that gut dysbiosis is associated with coronavirus disease 2019 (COVID-19) infection and may persist long after disease resolution. Patients with COVID-19 had significant alterations in fecal microbiomes compared with controls, characterized by the enrichment of opportunistic pathogens and depletion of beneficial commensals at the time of hospitalization and at all time points during hospitalization. Deficiency of symbionts and gut dysbiosis persisted even after the elimination of severe acute respiratory syndrome coronavirus 2 (SARS-CoV-2) and the resolution of respiratory symptoms [3]. It has been shown that microbiota diversity does not recover even 6 months after convalescence [4]. Moreover, the use of antimicrobials in patients with COVID-19, often excessive, can lead to additional destruction of the microbiota, as well as to the growth and spread of antimicrobial resistance (AMR), especially in intensive care units [5]. The problem of bacterial resistance to antibiotics encourages the search for alternative methods of limiting bacterial growth and restoring the normal balance of the microbiota in the human body.

The prevalence of bacteriophages in the biosphere and the regulatory role they play in various ecosystems, such as the oceans or the human gut, are of great interest to scientists and clinicians [6]. Bacteriophages are among the candidates as potential microbial modifiers to promote gut health [7]. Recently, researchers and clinicians have increasingly begun to work with bacteriophages to target antibiotic-resistant strains of bacteria [8]. It has been shown that the use of bacteriophages against an antibiotic-resistant bacterial population not only effectively limits bacterial growth but can also restore sensitivity to antimicrobials and reduce the number of antibiotic resistance genes in the population [9]. Such effects may be associated with switching the mechanisms of adaptation of the bacterial population to bacteriophage infection. Moreover, bacteriophages can encode exopolysaccharide depolymerases to facilitate penetration into biofilms and infection of resident bacteria [10]. Such bacteriophages can use exopolysaccharides as primary receptors, and sequential cleavage of polymer bonds without virus dissociation allows virions to make their way through the polysaccharide layer until they reach secondary receptors in the cell membrane, binding to which initiates subsequent stages of infection [11]. Phages demonstrate a high degree of host specificity, allowing them to be used for the selective reduction in pathogenic and/or etiologically significant bacteria in the microbial environment, including antibiotic resistance.

The relevance of the problem can be considered by the example of *Klebsiella pneumoniae*—one of the *Enterobacteria* species, which taxonomically refers to *Proteobacteria*. *K. pneumoniae* is a Gram-negative bacterium that colonizes human mucous membranes, mainly the gastrointestinal tract, less often the nasopharynx [12]. It has been shown that the intestinal predominance of *Proteobacteria* leads to a fivefold increased risk of bacteremia [13], suggesting that the bacterial density of colonizing strains plays a role in disease development. Medical manipulations such as gastroscopy, bronchoscopy, or artificial lung ventilation can introduce bacteria into new regions of the internal organs, where they can propagate actively and cause infections [14]. Moreover, damage to body tissues during manipulation can release additional nutrients to the bacteria and promote their active growth. *K. pneumoniae* is associated with human diseases such as pneumonia, urogenital infection, liver abscess, bloodstream infection, etc. It is one of the most frequently detected bacteria in human respiratory tract infections, especially in hospitalized patients with pneumonia associated with treatment in intensive care units (ICU), including ventilator-associated pneumonia (VAP) [15]. Among bacterial complications after viral respiratory tract infections, including SARS-CoV-2, pneumonia associated with *K. pneumoniae* is also the most common [16,17]. The treatment of such complications is challenged by increasing levels of bacterial resistance to antimicrobial drugs. An increase in the incidence of *K. pneumoniae* infections caused by carbapenem-resistant strains has been detected worldwide, which has resulted from the use of carbapenem-class antibiotics against extended-spectrum beta-lactamases (ESBLs)-producing strains of *K. pneumoniae* [18,19,20]. 

Global studies show that a significant proportion of nosocomial *K. pneumoniae* isolates exhibit ESBLs and carbapenemases activity [21]. The formation of biofilms negatively affects the effectiveness of therapy for infections associated with *K. pneumoniae* since the biofilm matrix not only physically protects the bacteria but also facilitates the transfer of mobile genetic elements responsible for antibiotic resistance, which increases microbial tolerance to antibiotics, bacterial persistence, and spread. Moreover, the ability to form biofilms on tubes and medical devices increases the risk of acute infections in patients, especially on prolonged artificial life support [22].

The conducted study of the application of the bacteriophage cocktail demonstrated the potential of bacteriophages to selectively reduce the number of target organisms without disrupting the gut community [23]. Unlike antibiotics, which can disrupt microbial communities by predisposing to dysbiosis or creating ecological niches for pathogens [24], bacteriophages represent a new way of selectively modifying the gut microbiota, affecting the gut environment without causing global disturbances that can lead to microbial dysbiosis.

In the present study, the bacteriophage cocktail for oral administration targeting multiple species of bacteria was used for the rehabilitation of 30 patients after COVID-19 and treatment with various antibiotics. Furthermore, the effectiveness of the bacteriophages against the clinical strain of *K. pneumoniae* was evaluated for the first time using real-time imaging on a 3D Cell Explorer microscope.

## 2. Materials and Methods

### 2.1. Bacteriophages

In the in vitro study, bacteriophages vB_KpnS_FZ10, vB_KpnP_FZ12, and vB_KpnM_FZ14 were tested, which had been previously isolated from sewage waters and were fully characterized in published research [25]. Their morphology was assessed using transmission electron microscopy, thermal and pH stability were evaluated, one-step-growth parameters, host adsorption rate, host range and phage-resistant forms generation rate were characterized, and the complete genome was sequenced and analyzed. The latent period was 30 min for vB_KpnS_FZ10, vB_KpnP_FZ12, and vB_KpnM_FZ14. The burst size was approximately 80 particles per bacterial cell for vB_KpnS_FZ10 and vB_KpnP_FZ12 and 120 particles per bacterial cell for vB_KpnM_FZ14. The complete genome sequences have been deposited in GenBank under the accession numbers MK521904, MK521905, and MK521906, respectively. Raw Illumina reads are available on NCBI SRA under accession numbers SRR10037530, SRR10037529, and SRR10037528, respectively. The associated BioProject accession number is RJNA562287.

In the in vivo study, two different bacteriophage cocktails were used. Both cocktails were developed by research and production center “MicroMir” (RPC “MicroMir”) and included phages from an in vitro study. 

The cocktail for oral administration included 43 phages active against clinical strains of *Enterococcus faecalis*, *Enterobacter cloacae*, *Enterobacter kobei*, *Proteus vulgaris*, *Proteus mirabilis*, *K. pneumoniae*, *Pseudomonas aeruginosa*, *Staphylococcus aureus*, *Staphylococcus warneri*, *Staphylococcus haemolyticus*, *Staphylococcus capitis*, *Staphylococcus caprae*, *Staphylococcus succinus*, *Enterococcus faecium*, and *Citrobacter freundii*. The preparation consisted of a sterile suspension of phage particles in a physiological solution. The titer of each bacteriophage was between 10^5^ and 10^6^ PFU/mL.

The cocktail for inhalation included 45 phages active against clinical strains of *Acinetobacter baumannii*, *Stenotrophomonas maltophilia*, *K. pneumoniae*, *K. pneumoniae subsp. ozanae*, *P. aeruginosa*, *S. aureus*, *Staphylococcus epidermidis*, *S. warneri*, *S. haemolyticus*, *S. capitis*, *S. caprae*, *S. succinus*, *Streptococcus pyogenes*, *Streptococcus agalactiae*. The preparation consisted of a sterile suspension of phage particles in a physiological solution. The titer of each bacteriophage was between 10^5^ and 10^6^ PFU/mL. The efficacy and safety of this bacteriophage cocktail in comparison with conventional antibiotic therapy have been studied previously in a published paper [26].

### 2.2. Bacterial Strains

For the in vitro study, the clinical strain Kl 315 of *K. pneumoniae* from the RPC “MicroMir” collection was selected. The strain was examined on a MALDI-TOF Microflex mass spectrometer (Bruker, Billerica, MA, USA) and with biochemical tests (MIKROLATEST, Erba Mannheim) with further analysis on Multiskan Ascent spectrophotometer (Thermo Scientific, Waltham, MA, USA) before adding to the collection. The same strain was chosen to examine the properties of the bacteriophages used in this study [25].

### 2.3. In Vitro Real-Time Phage Lysis Assay

Cells from an overnight culture were suspended in BHI broth to 10^9^ CFU/mL, and 1 mL of suspension was added to a glass-bottom Petri dish and incubated for 1 h 40 min at 37 °C in aerobic conditions. Then, the bacteriophage cocktail was added to the Petri dish at a final concentration of 10^9^ PFU/mL. The mixture was incubated for 3 h 10 min. During the whole incubation period, video recording of the growth and lysis of the bacterial culture was performed using a 3D Cell Explorer microscope (Nanolive, Tolochenaz, Switzerland).

### 2.4. Participants

Our prospective pilot study included subjects with the post-COVID syndrome (*n* = 30) receiving the complex program of rehabilitation at the Federal Research and Clinical Center of Intensive Care Medicine and Rehabilitology from December 2021 to August 2022. Participants were eligible if they had recovered from COVID-19 3 or more months before the start of the study, if they reported experiencing fatigue at a level that was not present prior to COVID-19, and if they were otherwise healthy.

The following inclusion criteria were applied:Post-COVID patients of moderate severity (incl. ICU stay), discharged from the hospital more than 3 months ago;Presence of pulmonary foci of consolidation and fibrosis on chest CT (CT 1-3);Rehabilitation routing score—less than 3 points;Negative polymerase chain reaction (PCR) result for SARS-CoV-2.Exclusion criteria:Temperature above 38 °C;Increased dyspnea (above 30/min);Increase in systolic blood pressure above 180 mmHg or a decrease below 90 mmHg.

The bacteriophage cocktail for oral application was administered per os 2 times a day for 14 days, in the morning and in the evening. The amount of ingested phage suspension was 10 mL.

The bacteriophage cocktail for inhalations was administered by inhalation through an ultrasonic nebulizer 2 times a day for 14 days, the duration of each inhalation was 15–20 min, and the amount of inhaled phage suspension was 5 mL. 

### 2.5. Sample Collection

One stool sample and one venous blood sample were collected from each subject on the day of admission (before bacteriophage therapy) and 14 days after the start of therapy. Stool samples were collected into a disposable sterile container. The containers were transported to the lab and prepared immediately. The time from sample collection to sample analysis did not exceed 12 h. DNA was extracted from the supernatant: 0.1 g of the stool sample was mixed with 800 μL of the isotonic solution and vortexed until homogeneity. The resulting mixture was centrifuged at 11,300× *g* for 30 s (MiniSpin, Eppendorf, Hamburg, Germany). Further analysis was carried out following the protocol provided by the assay manufacturer (Colonoflor-16 by AlphaLab, St. Petersburg, Russia). Blood was collected from a venous catheter into an anticoagulant-free test tube. Serum samples were obtained by centrifuging the blood at 1500× *g* for 10 min (ELMI CM-6M, Riga Latvia). Serum aliquots (500 μL) were poured into disposable Eppendorf tubes, frozen, and stored at −20 °C until further use.

### 2.6. Analysis of Gut Microbiota Taxonomic Abundance

The composition of the gut microbiota was analyzed using Colonoflor-16 kits (AlphaLab; Russia), which include reagents for DNA extraction, PCR primers specific for all bacterial DNA (total bacterial mass), and species-specific primers for 16 microbial species. Measurements were performed using a CFX 96 Real-Time PCR Detection System (BioRad, Hercules, CA, USA).

### 2.7. CT Scan Analysis

To analyze the results of lung CT scans, the method of automatic calculation of the volume of damaged lung tissue according to the type of ground glass using the software “Ground glass” (InfoRad 3.0 DICOM Vievew, Moscow, Russia) was used. Segmentation of the right and left lung and trachea was performed with a threshold of −250 HU, and areas of damage within the lung with density in the custom range (from −785 HU to 150 HU) were identified.

## 3. Results

### 3.1. Experimental Part

#### In Vitro Real-Time Phage Lysis Assay

In the growth phase of the culture, before adding the bacteriophage complex, an increase in the number of bacteria in the microscope field of view was observed (Figure 1). The first stages of biofilm formation were observed: adhesion on glass and the beginning of microcolony formation. A full video of the growth phase is provided in Appendix A. 

The bacteriophage cocktail was added 1 h 40 min after the start of the incubation and filming. Within 2 h 30 min after the addition of phages, there was a slight decrease in the number of bacteria in the field of view but formed bacterial microcolonies were still visible. Active reduction in the number of bacteria in the field of view started 2 h 30 min after the addition of the bacteriophage complex, and within 30 min, all microcolonies were disrupted (Figure 2). A full video of the bacterial lysis phase is provided in Appendix A. Additionally, a lysis of the bacterium adhered to the glass in the center of the microscope field of view was captured (Appendix A).

There was a decrease in the area covered by bacterial cells from 27.9% to 10.5% (Figure 3).

Further observation showed no continuation of lysis or elimination of the bacterial culture.

### 3.2. Clinical Phage Assessment

#### 3.2.1. Subjects Characteristics

The age range was from 19 to 82 years (median: 62 (53; 67) years); 16 study participants were female (53%), 14 were male (47%), and their body mass index (BMI) was 33 (26; 37). All subjects had evidence of lung involvement on CT scans ranging from 10% to 75% (Stage I—50%, II—31%, III—12, IV—7%) during acute COVID-19. Antibacterial therapy was administered to more than 25% of subjects during the underlying illness (third-generation cephalosporins, amoxicillin in combination with beta-lactamase inhibitors). The comparison group consisted of subjects (*n* = 8) also who had undergone COVID-19 in the past and received the complex program of rehabilitation in August 2022 but did not receive bacteriophage cocktails. The age range was from 21 to 77 years (median: 53 (44; 62) years); 7 study participants were female (87%), and 1 was male (13%).

As a result of the rehabilitation program involving phage therapy, the patients’ saturation increased, respiratory rate normalized, and heart rate decreased within the reference values (Table 1). A statistically significant increase in hemoglobin level and decrease in erythrocyte sedimentation rate (ESR) (Table 1) were registered in comparison with the control group (Appendix A). A trend toward normalization of the leukocyte level was observed. X-ray imaging assessment comparing the volume of lung tissue lesion “retrospectively/now” revealed the following results: residual and limited pneumofibrosis in 55% of patients; resolution of pneumonia from 5% to 20% (previously 60%) in 30% of cases, and resolution of pneumonia up to 40% (previously 80%) in 15% (Appendix A). An example of CT scan analysis using the “Ground Glass” software is presented in Appendix A.

In almost all subjects, biochemical parameters did not exceed the reference values (Table 2).

A statistically significant increase in the level of total protein was also revealed, which may indirectly indicate an improvement in absorption processes in the intestine against the background of the normalization of the state of the gut biocenosis. A decrease in the level of C-reactive protein (CRP) as a marker of inflammation (Δ 0.12 after versus Δ 0.6 before) should be noted. C-reactive protein levels were elevated in 4 of 24 (17%) patients at admission and in 2 (8%) at discharge (*p* = 0.894).

#### 3.2.2. Gut Dysbiosis

The microbiota was characterized by various markers of dysbiosis upon admission to rehabilitation in 100 % of the subjects: excessive bacterial growth (92%), increased concentration (lg CFU/g) of proinflammatory types of microorganisms (*Bacteroides fragilis group*, *Candida* spp., *S. aureus*, *Proteus* spp., *Enterococcus* spp., *Enterobacter* spp., *Citrobacter* spp.) exceeding the reference values by 1.5–2 times, and low levels of *Bacteroides thetaiotimicron*, *Akkermansia muciniphila* compared with the reference values (Table 3).

The composition of the gut microbiota included *Clostridium difficile and Clostridium perfringens*, *Fusobacterium nucleatum*, *and Parvimonas micra*, which are not found in significant concentration in a healthy gut. It also included a high ratio of *Bacteroides fragilis group/Faecalibacterium prausnitzii*, a sign of anaerobic imbalance, a condition characteristic of inflammatory bowel diseases such as ulcerative colitis and Crohn’s disease, and autoimmune pathology as an indicator of a disturbed state of local intestinal immunity.

A trend towards a decrease in the total bacterial mass was revealed after the bacteriophage therapy, as well as a decrease in the concentrations of proinflammatory microorganisms (lg CFU/g): *S. aureus*, *Enterococcus* spp., *Clostridium* spp., *Proteus* spp., *Citrobacter* spp.

After 14 days of application of the bacteriophage cocktail, a statistically significant decrease in the inflammatory ratio was revealed (Figure 4). 

A statistically significant positive relationship was found between the number of *Escherichia coli* and total bacterial mass (R = 0.42, *p* = 0.003), total bacterial mass and inflammatory coefficient (R = 0.48, *p* < 0.001), and the number of *Escherichia coli* and *Bacteroides* spp. (R = 0.43, *p* = 0.003). The number of bifidobacteria negatively correlated with the inflammatory coefficient (R = −0.32, *p* = 0.028) (Figure 5).

## 4. Discussion

### Clinical Part

Our results demonstrate the potential of the bacteriophage cocktail to modulate the gut microbiome. 

Previously, phages were thought to cause minimal changes in the composition and diversity of the gut microbiota in experimental studies compared with antibiotics [27]. Other studies using gnotobiotic mice have shown that exposure to phages leads to compositional changes in the murine gut microbiota [28,29]. The conducted study of the application of the bacteriophage cocktail targeting *E. coli* showed a decrease in the number of fecal *E. coli* loads. No significant changes in alpha and beta diversity parameters were observed, suggesting that the phages consumed did not disrupt the microbiota. However, an increase in the number of representatives of the butyrate-producing genus *Eubacterium* and a decrease in the proportion of taxa most closely related to *Clostridium perfringens* was observed. Short-chain fatty acid production, inflammatory markers, and lipid metabolism were virtually unchanged, but there was a small but significant decrease in circulating interleukin-4 (Il-4). Taken together, these data demonstrated the potential of bacteriophages to selectively reduce the number of target organisms without disrupting the gut community [23].

However, a clinical study demonstrated a virtually unchanged microbiota profile over 4 weeks of phage therapy when added to systemic antibiotics in one patient with *S. aureus* infection [30]. There are several publications describing the use of inhaled [31] or intravenous [32] bacteriophages in patients with pneumonia, bronchitis, and infectious endocarditis. All studies confirm the safety and declare the absence of adverse reactions when using bacteriophages for therapeutic purposes. Previously, in our pilot clinical trial, the safety of bacteriophages was evaluated for the first time in a group of patients with a chronic critical illness. In addition to the absence of local and general adverse events, an important aspect to note is that the effectiveness of the technique was confirmed by the treatment outcomes seen in the phage therapy group, which were not inferior to those in the group receiving conventional antibiotic therapy [26]. 

In the present study, a cocktail of bacteriophages for ingestion was used. Almost all subjects had levels of total bacterial mass above the reference values on admission (92%, 22/24) that decreased slightly by the time of discharge. The microbiota of the subjects on admission was characterized by the presence of proinflammatory types of microorganisms and low levels of anti-inflammatory microorganisms, as in patients with the post-COVID syndrome. These findings are consistent with previously identified microbiota disturbances after COVID-19 [33,34,35]. Dysbiosis was also confirmed by an increased ratio of the *Bacteroides fragilis/Faecalibacterium prausnitzii* group in 67 % of the subjects, which is a sign of anaerobic imbalance. Previously, similar changes in this ratio were observed in chronic critical illnesses, reaching maximum values during massive antibiotic therapy [36]. Application of the phage cocktails for two weeks did not cause adverse effects among the subjects, with normalization of stools and a reduction in the severity of gastrointestinal symptoms. 

After phage therapy, a statistically significant threefold decrease in the anaerobic imbalance ratio was detected. Although no correlation between this coefficient and C-reactive protein was found, there was a normalization of this biomarker and a statistically significant decrease in the ESR, a parameter that may also indicate inflammation. Reducing the intensity of the inflammatory response associated with bacterial infection, in particular, reducing the level of CRP, may be one of the effects of the bacteriophage application [37].

Bacteriophages in nature play an important role as regulators of bacterial populations, as well as in regulating the number of bacteria in the human gut microbiome. Understanding this role of bacteriophages became the basis for the development of the concept of immunity based on the adhesion of phages to mucus (bacteriophage adherence to mucus immunity) [38]. Bacteriophages on mucous membranes are considered the first line of counteraction to the development of dysbiosis; they respond to the bacterial proliferation process even before the reaction of cellular and humoral mechanisms of the immune system and contribute to faster achievement of a new point of balance by the microbiome in case of dysbiosis development [39].

We consider a significant clinical effect of bacteriophage inhalation to be a statistically significant increase in saturation and normalization of respiratory rate. The effectiveness of the inhaled form was confirmed by the results in patients with nosocomial pneumonia, where the treatment results were comparable in the conventional antibiotic therapy group [26].

For the in vitro study, we chose *K. pneumoniae*, a microorganism that frequently causes nosocomial pneumonia; moreover, patients often become infected with an intestinal colonization strain [40]. Because of the short incubation time, it was not possible to observe biofilm formation under the microscope. However, the first stages of biofilm formation—adhesion on glass and formation of microcolonies—were recorded. The characteristics of the bacteriophages studied in vitro have been previously examined: they encode polysaccharide-depolymerases, and the latent period is 30–35 min for all bacteriophages included in the cocktail. The delay in lysis on the video was due to the fact that the phage cocktail was added to the Petri dish laterally through a micro syringe in order to avoid significant mixing of the bacterial culture, and the filming took place in the central part of the Petri dish. Thus, the lysis of the bacteria began unevenly in the studied volume and reached the microscope field of view 2 h and 30 min after adding the bacteriophage complex. 

Active lysis lasted for 30 min, and no decrease in the number of bacteria in the field of view was observed during further incubation. This may be due to a complex population relationship between bacteriophages and bacteria. It is known that bacteriophages reproduce most actively in exponentially growing bacterial populations, while in a poor culture, lysis may stop, and bacteriophages may go into the so-called hibernation phase [41]. Additionally, bacteriophages usually actively lyse populations with high density. Thus, bacteriophages act as a regulator of bacterial populations in nature. This was observed in the present experiment when active lysis stopped after a significant reduction in the number of bacteria. The method allows us to visualize the growth of the bacterial population and the lysis of bacteria after the addition of bacteriophages. However, the closed type of the system does not allow sampling from it, so it is not possible to estimate lytic efficacy by cultural methods correctly. We propose to consider the analysis using the 3D Cell Explorer microscope as a method to visualize lysis in real time, which is rarely performed with phages because the more common method of imaging is electron microscopy, which involves fixation of the sample and staining.

This study had certain limitations. One of the limitations of this work is the use of real-time PCR to characterize the intestinal microbiota instead of 16s rRNA sequencing, which limits the taxa that can be detected and does not allow the evaluation of minor bacterial species. At the same time, this method provides the ability to track changes and identify major nosocomial pathogens virtually at the patient’s bedside in routine clinical practice. Another limitation was the subjects’ insufficient completion of questionnaires such as the Gastrointestinal Symptom Rating Scale (GSRS) [42] and the Bristol Stool Scale. Questionnaires were received from only four people, which does not allow us to discuss the significance of the obtained results. However, the four questionnaires received indicated a decrease in gastrointestinal symptoms according to GSRS and normalization of stool according to the Bristol Stool Scale. In the in vitro study, the main limitation was the choice of a clinical strain of *K. pneumoniae* for visualization. We chose *K. pneumoniae* because higher titers of this bacterium in the gut increase the risk of bacterial infections, including bacterial pneumonia. However, we did not find *Klebsiella* spp. exceeding the reference values in the intestines of patients, as was assumed when planning this work. Nevertheless, the objective of the in vitro experiment was not to show the efficacy of phages on a specific bacterium but to test a new method for visualizing the action of bacteriophages. Usually, methods used to visualize the action of bacteriophages do not allow for recording lysis in real time. In this study, we first imaged bacteriophage lysis in real time using a 3D Cell Explorer microscope.

## 5. Conclusions

In the present study, complex phage cocktails targeting several bacterial species were used for the first time. The results showed the safety of this application and the absence of side effects. The phage therapy group showed statistically significant improvements in saturation and respiratory rate as well as a decrease in inflammatory markers, such as ESR and *Bacteroides fragilis group/Faecalibacterium prausnitzii* ratio, according to PCR analysis of gut microbiota. The results demonstrate the relevance of bacteriophages in the rehabilitation of the microbiota of patients who have undergone COVID-19 and received antibiotic treatment. This work will serve as a starting point for a broader and more detailed study of the effects of phages on the human microbiome and the whole organism. 

## Figures and Tables

**Figure 1 viruses-14-02614-f001:**
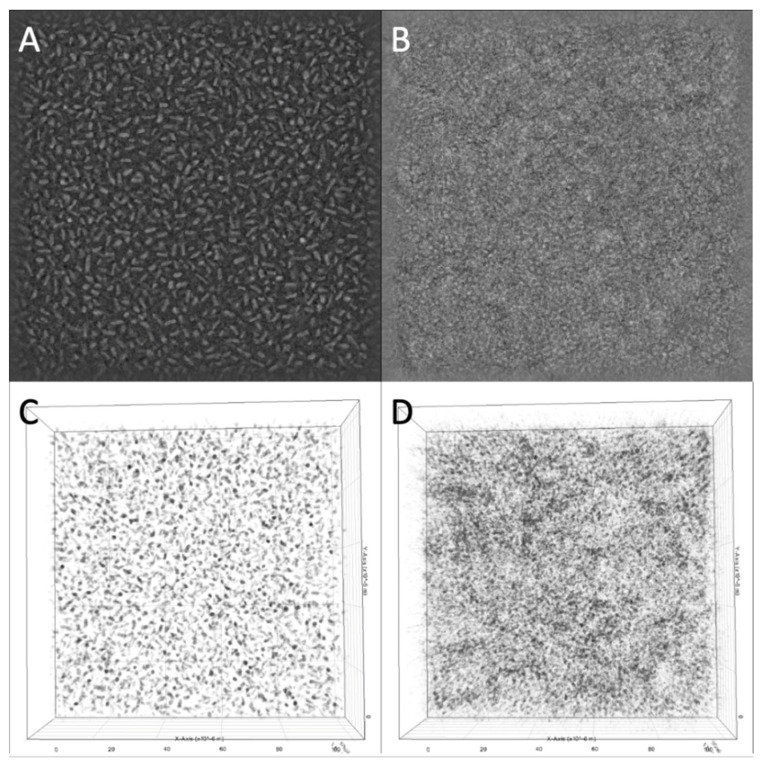
3D Cell Explorer micrographs of *K. pneumoniae* (Kl 315) culture: (**A**,**C**) in the start; (**B**,**D**) after 1 h 40 min of incubation. Magnification ×58, the field of view size is 100 × 100 µm. The experiment was conducted in one repetition.

**Figure 2 viruses-14-02614-f002:**
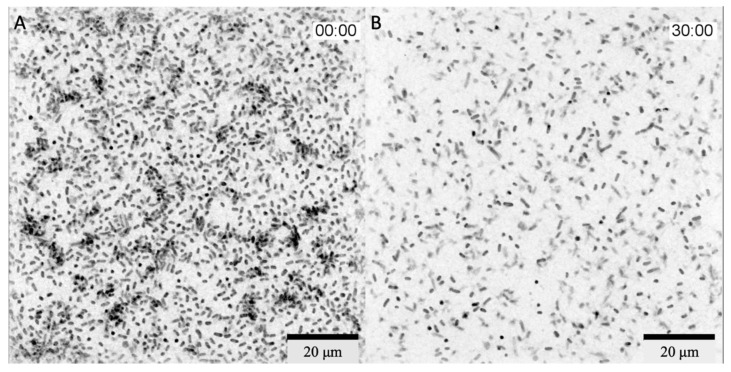
3D Cell Explorer micrographs of *K. pneumoniae* (Kl 315) culture: (**A**) 2 h 30 min; (**B**) 3 h after the addition of the bacteriophage cocktail. Magnification ×58, scale bar 20 µm. The experiment was conducted in one repetition.

**Figure 3 viruses-14-02614-f003:**
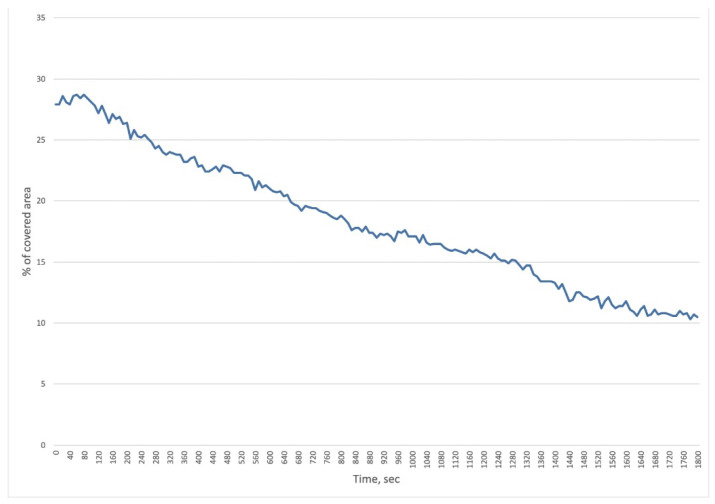
Reduction in bacterial coverage of the area observed with the 3D Cell Explorer microscope within 30 min (after 2 h 30 min from the addition of bacteriophage cocktail to the Kl 315 culture). The experiment was conducted in one repetition.

**Figure 4 viruses-14-02614-f004:**
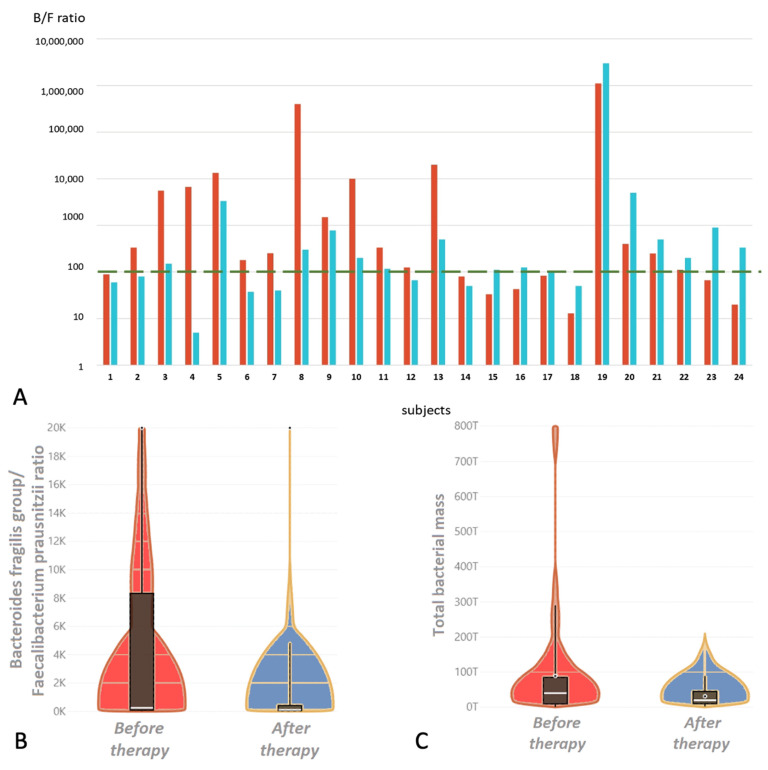
(**A**). *Bacteroides fragilis group/Faecalibacterium prausnitzii* ratio in subjects during therapy with bacteriophage cocktail; red bars before therapy, blue bars after. The green dotted line is the boundary of the reference value. Only 25% of the subjects had an increase in this indicator compared with the baseline. (**B**). Violin plot distribution of the *Bacteroides fragilis group/Faecalibacterium prausnitzii* ratio during therapy with bacteriophage cocktail; the red bar before therapy, the blue bar after. (**C**). Violin plot distribution of the total bacterial mass during therapy with bacteriophage cocktail; the red bar before therapy, the blue bar after.

**Figure 5 viruses-14-02614-f005:**
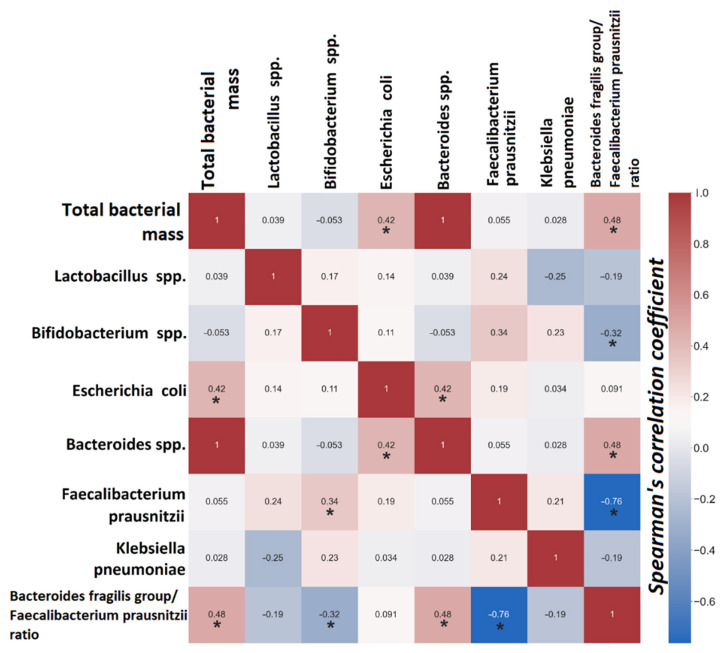
Heat map correlation relationships between microorganisms. Red squares are positive correlations, and blue are negative correlations. * Correlation is statistically significant.

**Table 1 viruses-14-02614-t001:** Clinical parameters in subjects on bacteriophage therapy (BPh). Data are presented as the median and interquartile range (IQR). * Correlation is statistically significant.

Parameters	Subjects, Median (Q1; Q3)		*p* Value
	Before BPh	After BPh	
**Saturation, percent**	96 (95; 96)	98 (98; 99)	*p* < 0.001 *
Respiratory rate (breaths per minute)	19 (18; 20)	16 (16; 16)	*p* < 0.001 *
Heart rate (beats per minute)	78 (74; 82)	72 (70; 72)	*p* < 0.001 *
Hemoglobin, g/L	135 (124; 150)	142 (132; 153)	0.007 *
White blood cells, 10^9^/L	6 (5; 7)	5 (5; 8)	0.436
Erythrocyte Sedimentation Rate (ESR), mm/hour	13 (8; 23)	9 (5; 13)	0.007 *

**Table 2 viruses-14-02614-t002:** Changes in the biochemical analysis of blood samples in subjects on the bacteriophage therapy (BPh). * Correlation is statistically significant.

Parameters	Reference Values	Subjects, Median (Q1; Q3)		*p* Value
		Before BPh	After BPh	
*Bilirubin*	5–21 µmol/L	15.6 (11.4; 18.8)	12.5 (11.2; 18.5)	0.058
*Total protein*	66–83 g/L	69.1 (66.3; 71.5)	71.2 (68.4; 72.6)	0.038 *
*Creatinine*	74–110 µmol/L	82 (74; 103)	89 (80; 98)	0.657
*Glucose*	4.1–5.9 mmol/L	5.7 (5.1; 5.9)	5.6 (5.2; 6.0)	0.236
*Cholesterol*	0–5.2 mmol/L	5.5 (4.5; 7.1)	5.9 (4.5; 6.2)	0.154
*Lactate dehydrogenase*	0–247 u/L	198 (168; 223)	197 (168; 229)	0.137
*Alanine aminotransferase*	0–50 u/L	21 (17; 32)	20 (15; 31)	0.679
*Aspartate aminotransferase*	0–50 u/L	23 (19; 32)	24 (19; 29.5)	0.225
*C-reactive protein (CRP)*	0–5 mg/L	0.6 (0.1; 0.9)	0.12 (0.1; 0.6)	0.061

**Table 3 viruses-14-02614-t003:** Taxonomic abundance of the gut microbiota in subjects before and after the bacteriophage therapy (BPh). * Correlation is statistically significant.

Parameters	Reference Values	Subjects, Median (Q1; Q3)		Indicator of Changes	*p* Value
		Before BPh	After BPh		
Total bacterial mass	<10^12^	4 × 10^13^ (9 × 10^12^; 9 × 10^13^)	2 × 10^13^ (10^13^; 6 × 10^13^)	⇩	0.171
*Lactobacillus* spp.	10^7^–10^8^	4 × 10^7^ (7 × 10^5^; 5 × 10^8^)	5 × 10^7^ (8 × 10^6^; 10^8^)	⇧	0.951
*Bifidobacterium* spp.	10^9^ –10^10^	3 × 10^10^ (4 × 10^9^; 7 × 10^10^)	2 × 10^10^ (3 × 10^9^; 6 × 10^10^)	⇩	0.535
*Escherichia coli*	10^7^ –10^8^	2 × 10^8^ (10^7^; 10^9^)	6 × 10^7^ (3 × 10^7^; 6 × 10^8^)	⇩	0.394
*Bacteroides* spp.	10^9^ –10^12^	4 × 10^13^ (9 × 10^12^; 9 × 10^13^)	2 × 10^13^ (10^13^; 6 × 10^13^)	⇩	0.171
*Faecalibacterium prausnitzii*	10^8^ –10^11^	9 × 10^10^ (9 × 10^9^;4 × 10^11^)	2 × 10^11^ (2 × 10^10^; 6 × 10^11^)	⇧	0.122
*Bacteroides thetaiotaomicron*	10^9^ –10^12^	5 × 10^9^ (4 × 10^8^; 2 × 10^10^)	2 × 10^9^ (7 × 10^8^; 10^10^)	⇩	0.327
*Akkermansia muciniphila*	<10^11^	2 × 10^7^ (10^7^; 10^12^)	5 × 10^6^ (2 × 10^5^; 5 × 10^8^)	⇩	0.93
*Enterococcus* spp.*	<10^8^	4 × 10^12^	5 × 10^5^		-
*Escherichia coli enteropathogenic*	<10^4^	-	-	-	-
*Klebsiella pneumoniae*	<10^4^	<10^5^	<10^5^	-	-
*Klebsiella oxytoca*	<10^4^	-	-	-	-
*Candida* spp.	<10^4^	2 × 10^6^ (3 × 10^5^; 2 × 10^7^)	2 × 10^6^ (3 × 10^5^; 3 × 10^6^)	-	0.23
*Staphylococcus aureus*	<10^4^	2 × 10^7^ (8 × 10^5^; 4 × 10^7^)	7 × 10^5^ (10^5^; 2 × 10^6^)	⇩	0.999
*Clostridium difficile* *	-	3 × 10^7^	10^5^	⇩	-
*Clostridium perfringens*	-	3 × 10^6^ (2 × 10^6^; 10^7^)	3 × 10^5^ (10^5^; 3 × 10^6^)	⇩	0.23
*Proteus vulgaris/mirabilis*	<10^4^	10^7^ (10^7^; 10^7^)	2 × 10^6^ (4 × 10^5^; 5 × 10^6^)	⇩	0.18
*Citrobacter* spp.	<10^4^	10^8^ (2 × 10^5^; 5 × 10^13^)	-	⇩	-
*Enterobacter* spp.	<10^4^	3 × 10^6^ (8 × 10^5^; 3 × 10^7^)	10^7^ (4 × 10^6^;10^8^)	⇧	0.53
*Fusobacterium nucleatum*	-	4.5 × 10^5^ (10^5^; 7 × 10^5^)	4 × 10^5^ (10^5^; 7 × 10^5^)	-	1.00
*Parvimonas micra*	-	3 × 10^5^ (10^5^; 4 × 10^5^)	9 × 10^5^ (6 × 10^5^; 10^6^)	-	0.19
*Bacteroides fragilis group/Faecalibacterium prausnitzii* Ratio	0.1–100	252 (80; 10,000)	111 (50; 500)	⇩	0.033 *

## Data Availability

Not applicable.

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
