# Peer review of "Bacteriophage Cocktails in the Post-COVID Rehabilitation"

_viruses, 2022, doi:10.3390/v14122614_

Round 1

Reviewer 1 Report (New Reviewer)

The authors have improved the manuscript a lot. Some minor type mistakes are needed to be revised before acceptance. For example, the genus name of bacteria is  abbreviate after they have been used once.

Author Response

Point 1: The authors have improved the manuscript a lot. Some minor type mistakes are needed to be revised before acceptance. For example, the genus name of bacteria is abbreviate after they have been used once.

Response 1: Abbreviated everywhere in the text except the tables to make them more understandable for those who will look at them separately from the text of the article.

Reviewer 2 Report (Previous Reviewer 3)

The revised manuscript from Zurabov et al examines the effect of bacteriophage administration on clinical parameters and gut microbiome in COVID-19 patients. While the revisions have improved the quality of the manuscript, a few outstanding questions/comments remain:

1. Please add the percent saturation, respiratory rate, and heart rate of the control group subjects to Table S1.

2. The addition of the CT data is very useful and should be expanded upon. Please add the CT lung volume assessment to Table 1 rather than just describing the results in the text. Do the authors have representative images of the CT findings they can include? 

Author Response

Point 1. Please add the percent saturation, respiratory rate, and heart rate of the control group subjects to Table S1.

Response 1: Added control group data to Table S1.

Point 2. The addition of the CT data is very useful and should be expanded upon. Please add the CT lung volume assessment to Table 1 rather than just describing the results in the text. Do the authors have representative images of the CT findings they can include? 

Response 2: Added CT lung volume assessment to Supplementary materials (Table S2). Added a Figure S1 to Supplementary materials as an example of CT data analysis.

Reviewer 3 Report (Previous Reviewer 1)

Dear Authors

I have carefully read your manuscript and have the following queries:

1. Klebsiella pneumonia should be written as Klebsiella pneumoniae, Check throughout the manuscript. 

2.Sars-Cov-2 should be written as SARS-CoV-2.

3. I have major concerns about ethical permissions, where these were obtained and mentioned?

4.  Demographic characteristics should be a part of results, not methods.

5. Authors should declare informed consent of the participants. 

6. Figure 2: MKM??

7. Conclusions are not robust to comprehend easily. 

Author Response

Point 1: Klebsiella pneumonia should be written as Klebsiella pneumoniae, Check throughout the manuscript. 

Response 1: Revised.

Point 2: Sars-Cov-2 should be written as SARS-CoV-2.

Response 2: Revised.

Point 3: I have major concerns about ethical permissions, where these were obtained and mentioned?

Response 3: Yes, the study was conducted in accordance with the Declaration of Helsinki and approved by the Ethics Committee of Federal Research and Clinical Center of Intensive Care Medicine and Rehabilitology (protocol code РР #4/20 from 22.09.2020) and informed consent was obtained from all subjects involved in the study. We have previously mentioned it in the Institutional Review Board Statement (Ln 485-487 in the revised manuscript,  Ln 471-473 in the original version) and Informed Consent Statement (Ln 488-489 in the revised manuscript, Ln 474-475 in the original version). If necessary, we can provide an informed consent form.

Point 4:  Demographic characteristics should be a part of results, not methods.

Response 4: Moved this paragraph to the results section.

Point 5: Authors should declare informed consent of the participants.

Response 5: We have previously declared it in the Informed Consent Statement (Ln 488-489 in the revised manuscript, Ln 474-475 in the original version). If necessary, we can provide an informed consent form.

Point 6: Figure 2: MKM??

Response 6: Changed to “μm”.

Point 7: Conclusions are not robust to comprehend easily.

Response 7: Modified.

Round 2

Reviewer 3 Report (Previous Reviewer 1)

The manuscript can be accepted after spell/grammar checks. 

This manuscript is a resubmission of an earlier submission. The following is a list of the peer review reports and author responses from that submission.

Round 1

Reviewer 1 Report

Dear Authors

I have carefully reviewed your manuscript but I have a major concern about the design of the study. The authors have not established the efficacy of bacteriophage therapy in treating infected patients. The authors have established the clinical outcomes of patients after phage administration. Authors have also established gut dysbiosis in only one subject which makes data more vulnerable. In conclusion, it is difficult to comprehend the outcomes of the present study. 

Thank you. 

Reviewer 2 Report

This manuscript investigated the effect of a bacteriophage cocktail in post COVID patients (n=24) to explore the hypothesis that phage administration would restore the normal balance of their gut microbiota. The authors claim that the phage application was successful in reducing the severity of gastrointestinal symptoms and in the normalization of stools. The authors also show the effect of three Klebsiella phages on in vitro culture of a K. pneumoniae clinical strain using real-time visualization on a 3D Cell Explorer microscope. The research is important because bacteriophages are among the candidates to promote gut health and are important to treat infections caused by extensively-drug resistant bacteria. However, the methods and approaches used in this work as well as the results presented need better explanation and additional clarification. I have some concerns, which must be addressed before considering the manuscript for publication.

 1)    The authors claim that the effect of three previously characterized phages against K. pneumoniae was evaluated for the first time using real time visualization on a 3D Cell Explorer microscope. How this method can be compared to classical methods to assess phage lytic efficacy?

2)    Provide in the legend of Figures 1 and 2 the MOI (multiplicity of infection) used in the assay. Were the 3 phages in equal proportion in the cocktail used in the in vitro assay?

3)    I strongly recommend that Figures 1, 2 and 3 are combined in a single Figure. Moreover, it should be mentioned that this in vitro assay was performed only once, as I understood.

4)    Provide CFU/mL change in the in vitro assay. Were the cells in the image viable after treatment with 3h phage cocktail?

5)    Provide more details on the phage cocktail developed by RPC “MicroMir” for oral administration. Phage particle number/mL? Phage suspension vehicle? What was the phage cocktail dosing defined per patient?

6)    Clarify the statement in Ln 128-130. It was not clear if this approach was applied in this work. If this was not the case, reword this sentence.

 Specific comments:

1)    Ln 15-17: Consider revising this sentence.

2)    Ln 75: Biotopes? Please clarify or change the word.

3)    Ln 77: “serve as an additional source of nutrition”?.  Please clarify or modify the sentence

4)    Ln 112 -114: Clarify that these phages have been previously characterized in a published work (reference #25).

5)    Ln 131: 3.1 subtitle should be modified (In vitro phage lysis assay?)

6)    Ln 214: 3.2 subtitle should be modified (Clinical phage assessment?)

7)    Ln 293: Clarify the meaning of “normalization of stools”.

Reviewer 3 Report

This manuscript by Zurabov et al describes the development of a bacteriophage cocktail to treat dysbiosis commonly seen following recovery from COVID-19. The authors demonstrate that the bacteriophages effectively target Klebsiella pneumoniae in vitro. They then report the safety and efficacy profile of bacteriophage therapy to 24 patients who have recovered from COVID-19. While the data is intriguing, there are several areas of suggested improvement for the authors to consider:

Major Points:

1. The largest issue with the data presented is the lack of a control group that was administered a vehicle solution. While paired pretreatment and posttreatment data are critical, a control group should have also been included. This is especially important to determine whether the resolution of gastrointestinal symptoms and improvement of bilirubin levels were due to the bacteriophage treatment and not instead a natural resolution.

2. Most of the in vitro work centers around the bacteriophages’ efficacy against Klebsiella pneumoniae. However, Klebsiella spp. were not clinically relevant levels in the pretreatment stool samples, questioning the clinical relevance of this data.

3. The abstract mentions normalization of stools and resolution of gastrointestinal symptoms of patients, but no data is presented supporting these statements. How was this assessed? Was a clinical scoring or stool quality system used? Were patients given a questionnaire to fill out?

Minor Points:

1. More detail is needed regarding the administration of the bacteriophage cocktail in the Materials and Methods.

2. In Table 1, a clear indicator of the relative change of different bacterial species in pretreatment versus posttreatment stool samples would aid in ease of interpretation of the results.